# Role of Gut Microbial Metabolites in Ischemic and Non-Ischemic Heart Failure

**DOI:** 10.3390/ijms26052242

**Published:** 2025-03-02

**Authors:** Mohammad Reza Hatamnejad, Lejla Medzikovic, Ateyeh Dehghanitafti, Bita Rahman, Arjun Vadgama, Mansoureh Eghbali

**Affiliations:** Division of Molecular Medicine, Department of Anesthesiology and Perioperative Medicine, David Geffen School of Medicine, University of California Los Angeles, BH-550 CHS, Los Angeles, CA 90095-7115, USA; mhatamnejad@mednet.ucla.edu (M.R.H.); lmedzikovic@g.ucla.edu (L.M.); atiyehd92@gmail.com (A.D.); bitanicole.r@gmail.com (B.R.); avadgama03@gmail.com (A.V.)

**Keywords:** heart failure, gastrointestinal microbiome, gut dysbiosis, trimethylamine N-oxide, short-chain fatty acids, bile acids, probiotics, prebiotics, fecal microbiota transplantation, hydrogen sulfide

## Abstract

The effect of the gut microbiota extends beyond their habitant place from the gastrointestinal tract to distant organs, including the cardiovascular system. Research interest in the relationship between the heart and the gut microbiota has recently been emerging. The gut microbiota secretes metabolites, including Trimethylamine N-oxide (TMAO), short-chain fatty acids (SCFAs), bile acids (BAs), indole propionic acid (IPA), hydrogen sulfide (H_2_S), and phenylacetylglutamine (PAGln). In this review, we explore the accumulating evidence on the role of these secreted microbiota metabolites in the pathophysiology of ischemic and non-ischemic heart failure (HF) by summarizing current knowledge from clinical studies and experimental models. Elevated TMAO contributes to non-ischemic HF through TGF-ß/Smad signaling-mediated myocardial hypertrophy and fibrosis, impairments of mitochondrial energy production, DNA methylation pattern change, and intracellular calcium transport. Also, high-level TMAO can promote ischemic HF via inflammation, histone methylation-mediated vascular fibrosis, platelet hyperactivity, and thrombosis, as well as cholesterol accumulation and the activation of MAPK signaling. Reduced SCFAs upregulate Egr-1 protein, T-cell myocardial infiltration, and HDAC 5 and 6 activities, leading to non-ischemic HF, while reactive oxygen species production and the hyperactivation of caveolin-ACE axis result in ischemic HF. An altered BAs level worsens contractility, opens mitochondrial permeability transition pores inducing apoptosis, and enhances cholesterol accumulation, eventually exacerbating ischemic and non-ischemic HF. IPA, through the inhibition of nicotinamide N-methyl transferase expression and increased nicotinamide, NAD+/NADH, and SIRT3 levels, can ameliorate non-ischemic HF; meanwhile, H_2_S by suppressing Nox4 expression and mitochondrial ROS production by stimulating the PI3K/AKT pathway can also protect against non-ischemic HF. Furthermore, PAGln can affect sarcomere shortening ability and myocyte contraction. This emerging field of research opens new avenues for HF therapies by restoring gut microbiota through dietary interventions, prebiotics, probiotics, or fecal microbiota transplantation and as such normalizing circulating levels of TMAO, SCFA, BAs, IPA, H_2_S, and PAGln.

## 1. Introduction

The gut is part of the gastrointestinal system and encompasses a diverse collection of microorganisms [1]. Gut-habitant commensal bacteria, i.e., gut microbiota, not only participate in the digestion of nutritional products but also synthesize and release metabolites to modulate host metabolism [2,3]. A plethora of triggering factors can change the normal gut microbiota, known as gut dysbiosis. Accordingly, host–gut microbiota interaction and dysbiosis have been recognized in the pathophysiology of non-gastrointestinal conditions, such as immune disorders [4], obesity [5], and diabetes [6].

Cardiovascular diseases (CVD) are the leading cause of mortality and morbidity worldwide [7], and knowledge of the role of gut microbiota and dysbiosis in different CVD has recently been emerging [8,9,10]. Heart failure (HF), resulting from either ischemic insult, such as myocardial infarction (MI) or myocardial ischemia-reperfusion injury (IRI), or non-ischemic insults, such as hypertension that leads to cardiac hypertrophy and fibrosis, has recently attracted research interest concerning the gut microbiota.

The gut microbiota exerts its systemic effects through the secretion of well-known metabolites such as trimethylamine N-oxide (TMAO), short-chain fatty acids (SCFAs), and bile acids (BAs), along with less-studied metabolites including indole propionic acid (IPA), hydrogen sulfide (H_2_S), and phenylacetylglutamine (PAGln). Each of these gut microbiota metabolites is released into the circulation and may subsequently affect the heart; this is known as the gut–heart axis (Figure 1). In HF patients, in addition to dysbiosis, increased circulating microbiota metabolites have been reported compared to healthy controls [11,12]. The present review explores the roles of the secreted microbiota metabolites in the pathophysiology of ischemic and non-ischemic HF by summarizing current knowledge from clinical studies and experimental models.

## 2. The Gut Microbiota Biochemical Factory: Generation of Metabolites

Consumed foods will turn into chyme in the stomach, pass to the duodenum, mix with the bile, and the nutrients eventually reach the small or large intestine. The gut microbiota converts the micronutrients to TMAO, SCFAs, secondary BAs, IPA, H_2_S, and PAGln which are reabsorbed into the portal vein and released into the systemic circulation (Figure 1).

### 2.1. TMAO

Under physiological conditions in the metabolism, nine recognized human intestinal bacterial strains [13] can convert choline, betaine, L-carnitine, lecithin, trimethyl lysine, γ-butyrobetaine, phosphatidylcholine, and glycerophosphocholine, into trimethylamine (TMA) and release it into the bloodstream [14,15]. Four different enzymes have been identified to facilitate this process: choline-TMA lyase [16], carnitine monooxygenase [17], betaine reductase [18], and TMAO reductase [19]. Circulatory TMA will be captured in the liver and oxidized into TMAO by hepatic flavin-containing monooxygenase (FMO) [20]. For this process, five functional enzymes of the FMO family exist; however, the rate-limiting enzyme FMO3 is the most efficient enzyme for TMA conversion [21]. After acting on targeted organs, TMAO will be cleared by the kidney [22] and reduced to TMA in the gut by bacterial TMAO reductase [23]. Any disruption in normal metabolism, including the decreased clearance of TMAO due to renal impairment [24] or increased production of TMAO in gut dysbiosis [13], leads to tissue accumulation of TMAO (Figure 1).

### 2.2. SCFAs

SCFAs, known as carboxylic acids, are naturally produced in small amounts in the liver and significant amounts in the colon, where the colonic bacteria ferment the fibers and resistant starches [25]. Acetate, propionate, butyrate, valerate, and isovalerate are the main types of SCFAs produced in different quantities; as such, acetate is the most abundant, whereas valerate and isovalerate are the least abundant [26]. Also, succinate serves as a key intermediate metabolite produced during the fermentation process by gut microbes, where indigestible dietary fibers and host-derived carbohydrates are broken down into SCFAs [27]. Butyrate, valerate, and isovalerate are mainly utilized by colonocytes as the energy source [28], while acetate and propionate reach the liver through the portal vein. Hepatocytes subsequently metabolize propionate, while acetate either remains in the liver or is released systemically to the peripheral venous system [29]. Propionate and acetate in the liver serve as substrates for the energy-producing tricarboxylic acid cycle and are efficiently metabolized to produce glucose [30]. Even though most SCFAs are used by colonocytes or hepatocytes locally, a small amount of SCFAs enter the bloodstream. SCFAs are recognized not only for their local anti-inflammatory properties by modulating immune cell chemotaxis, reactive oxygen species (ROS) production, and cytokine release but also for their crucial role in preventing systemic inflammation [31]. Another role of gut microbiota-derived SCFAs is that they act as an energy source for enterocyte tight junction homeostasis [32,33,34,35]. As such, dysbiosis leads to leaky gastrointestinal epithelium, allowing the translocation of luminal endotoxins and commencing a systematic inflammation which affects various organs, including the cardiovascular system (Figure 1) [36].

### 2.3. Bile Acids

The liver generates primary BAs from cholesterol catabolism, with a rate-limiting conversion operated by cytochrome P450 enzyme CYP7A1. Two primary BAs, cholic acid and chenodeoxycholic acid, undergo conjugation with the amino acids taurine or glycine [37]. These products are condensed in the gallbladder and then secreted into the duodenum, undergoing a microbiota-mediated conversion into secondary BAs in the jejunum. Next, secondary BAs are reabsorbed by the transporters in the ilium and return to the liver by the portal vein [38]. This efficient enterohepatic circulation recycles 95% of BAs [39]. However, the remaining BAs may circulate through the systemic bloodstream and interact with BA receptors in tissues (Figure 1) [40]. Traditionally, BAs have been known as lipid-emulsifying effectors which aid with lipogenic digestion and absorption [41]. However, the mechanistic role of BAs in different physiological processes [42], such as glucose homeostasis [43] and thyroid function [44], have been elucidated. BAs are signaling molecules that affect cardiac function by binding to the Farnesoid X receptor (FXR) and Takeda G Protein-Coupled receptor-5 on cardiomyocytes [37].

### 2.4. Other Metabolites

In addition to the widely recognized microbial metabolites such as TMAO, SCFAs, and BAs, the gut microbiome can also produce and release other compounds, including IPA, H_2_S, and PAGln, through distinct metabolic pathways (Figure 1) [45]. Bacteria can convert tryptophan from protein-rich foods into IPA [45] which not only enhances local intestinal barrier integrity and prevents dysbiosis [46], but also can affect molecular salvage pathways [47]. Both colonocytes and gut microbiota produces H_2_S through sulfate reduction or cysteine catabolism via the enzymatic action of cysteine desulfhydrase [48,49]. Microbiota-derived H_2_S regulates gut inflammation, supports tissue repair, and acts as a gasotransmitter to promote vasodilation and systemic effects [50,51]. Following the consumption of protein-rich foods containing phenylalanine, gut bacteria metabolize phenylalanine into phenylacetic acid [52]. This metabolite is absorbed into the portal vein and, upon reaching the liver, undergoes conjugation with glutamine, leading to the formation of PAGln, which then enters systemic circulation [52]. PAGln has been shown as biomarker for cardiovascular and cerebrovascular accidents [53,54].

## 3. Pathophysiological Insight in Ischemic and Non-Ischemic Heart Failure

Distinct pathophysiologies underlie the development of ischemic and non-ischemic HF. MI arises due to the obstruction of coronary arteries by atherosclerotic plaque or thrombus formation, resulting in myocardial ischemia [55]. The timely restoration of coronary blood flow to the ischemic myocardium remains the foundation of clinical therapy. Although reperfusion is necessary to reestablish oxygen and nutrient delivery and salvage ischemic myocardium, it paradoxically exacerbates the injury, a phenomenon known as reperfusion injury [9]. After MI, the heart undergoes adverse ventricular remodeling, including cardiomyocyte death and replacement fibrosis, which, over time, leads to HF [11,12]. Key pathophysiological processes of IRI include calcium accumulation in mitochondria leading to the opening of the mitochondrial permeability transition pores (mPTP), and mitochondrial swelling and cytochrome C release causing the apoptosis of cardiomyocytes [56]. Additionally, damaged mitochondria produce excess ROS. Injured and dying cardiomyocytes release their cellular contents, which in turn trigger a pro-inflammatory response consisting of cytokine production and pro-inflammatory cell recruitment [57]. Finally, fibroblasts proliferate and differentiate into myofibroblasts to promote fibrotic scar formation to maintain the structural integrity of the infarcted heart [57]. Also, endothelial–myofibroblast transition resulting in coronary artery fibrosis can participate in ischemic HF [58].

Non-ischemic heart failure occurs in the absence of vascular blockages when the myocardium loses its ability to effectively contract and eject the blood, known as HF with reduced ejection fraction (HFrEF). Non-ischemic HF is often preceded by cardiac hypertrophy [13]. While cardiac hypertrophy is initially compensatory, it often progresses to adverse hypertrophy with fibrosis and ultimately to HF when insults, such as hypertension, persist. Key pathophysiological pathways in non-ischemic HF include mitochondrial dysfunction [59] and the disruption of intracellular calcium homeostasis in cardiomyocytes [59,60], cardiac fibroblast-to-myofibroblast transition due to enhanced TGF-ß-SMAD signaling [61,62], as well as inflammatory responses in the myocardium [63].

Recently, the involvement of epigenetic processes, such as DNA methylation, histone modifications, and non-coding RNA regulation, in HF has been emerging [64]. The disruption of DNA methylation and histone modifications can interfere with crucial pathways essential for cardiomyocyte viability, including mitochondrial activity, regulation of oxidative stress, and inflammatory responses [65]. Also, non-coding RNA, including microRNAs, long non-coding RNAs, and circular RNAs, participates in cardiac fibrosis, hypertrophy, and remodeling [66].

## 4. Gut Dysbiosis in HF Patients and Experimental Models

Several studies have revealed altered gut microbiota in HF patients. Indeed, β-diversity analysis indicated significant differences in microbial composition among HFrEF and controls [67]. A shift from microbiota with anti-inflammatory properties toward bacteria with pro-inflammatory features was observed by performing stool microarray analyses in HF patients compared to healthy subjects [10]. Indeed, several studies have shown a shift toward the predominance of bacteria usually attributed to a pro-inflammatory role in stool of HF patients, such as the enhancement of *Streptococcus* and *Veillonella* spp. [68], lower *Firmicutes* to *Bacteroidetes* ratio [69], reduction of *Ruminococcaceae* spp. [9], depletion in the *Eubacterium rectale* and *Dorea long catena* genus [70], and dominance of pathogenic bacteria spp., including *Campylobacter*, *Shigella*, *Salmonella*, and *Yersinia* [71].

Experimental animal models have verified the role of gut dysbiosis in HF. Gut microbiota diversity decreased drastically in rats with HF in the isoproterenol-induced cardiac hypertrophy model compared to healthy rats [72]. The overgrowth of bacteria involved in pro-inflammatory responses, such as *Prevotella*, and the reduction of anti-inflammatory microbiota, including *Roseburia*, *Lactobacillus*, and *Butyrivibrio*, was observed in mice with HF [72]. In addition, gut dysbiosis and decreased microbiota diversity (a higher ratio of *Firmicutes* to *Bacteroidetes* and a decrease in abundance of *Muribaculaceae*, *Lachnospiraceae*, and *Lactobacillaceae*) were observed in HF rats on a high-salt diet with hypertension [73]. In obese ZSF1 rats with hypertension, diabetes, and HF, fecal analysis showed that *Lactobacillaceae*, *Ruminococcaceae*, *Erysipelotrichaceae*, and *Lachnospiraceae* were significantly increased compared to healthy lean rats [74].

Worth mentioning is that a bidirectional connection exists between HF and dysbiosis [75]. Peripheral tissue hypoperfusion due to ventricular failure is common in end-stage HF [76]. The gastrointestinal system, one of the most susceptible organs to ischemia, will undergo cell hypoxia, anaerobic metabolism, and lumen pH decrement, which eventually resulting in dysbiosis [77].

## 5. Gut Microbiota Metabolites as Primers for Ischemic and Non-Ischemic Heart Failure

In addition to changes in gut microbiome composition, changes in specific microbiota metabolites, particularly TMAO, SCFAs, and BAs, and some less-studied metabolites including IPA, H_2_S, and PAGln, have been reported in HF patients and experimental HF models (Table 1).

### 5.1. TMAO

#### 5.1.1. Ischemic Heart Failure

Increased levels of TMAO have been reported in the plasma of patients suffering from myocardial ischemia, and high levels of TMAO were positively correlated with major advanced cardiac events [78]. In a cross-sectional study of patients undergoing cardiac surgery, TMAO levels were positively correlated to the number of infarcted coronary vessels [79]. In a cohort study, plasma TMAO levels were significantly elevated in both ischemic and non-ischemic HF patients compared to healthy controls, with the highest TMAO levels observed in patients with ischemic HF. Interestingly, TMAO was positively associated with NYHA functional class, MI, and major advanced cardiac events [11]. In a cohort of ischemic HF patients, a significant increase in serum TMAO was a strong predictor of adverse outcomes [80]. Additionally, high levels of TMAO, together with NT-pro B-type natriuretic peptide (BNP), have been shown to be a valuable prediction tool panel for risk stratification of ischemic HF patients [81]. Comparing the serological concentration of bacterial metabolites between young acute MI patients with healthy controls revealed that TMAO and acetate levels at hospital admission were significantly higher in patients with MI. After three months, follow-up evaluation showed that all metabolites except butyrate returned to their corresponding level of the healthy controls [82].

Increased TMAO plasma levels have also been demonstrated in experimental models of myocardial ischemia induced by the permanent ligation of LAD for several weeks or short-term ligation of LAD (~30 min) followed by reperfusion known as IRI (Figure 2A and Table 1). In a myocardial ischemia model induced by the permanent ligation of LAD for 6 weeks in rats, Yang et al. have demonstrated that Luhong, a traditional Chinese anti-inflammatory and antioxidant medicine, attenuated LV remodeling and concomitantly stool sequencing revealed decreased TMAO-producing bacteria such as *Bacteroidales, Alistipes*, and *Phascolarctobacterium* in Luhong-treated rats. As such, it was suggested that Luhong attenuates ischemia-induced cardiac remodeling by mitigating gut dysbiosis [83]. In a similar experimental setting, Weng et al. demonstrated the protective role of Buyang Huanwu Decoction, another traditional Chinese immunomodulatory medicine, against ischemic HF. BHD-treated rats who underwent LAD ligation had lower levels of serum TMAO and Tei index (the ratio of ventricular isovolumic contraction time and isovolumic diastolic time to ejection time), indicating that TMAO may mediate ischemic HF pathophysiology [84]. In a similar HF study induced by 4 weeks of permanent LAD ligation, rats treated with traditional Chinese medicine Yixintai exhibited reductions in plasma TMAO concomitantly with lower expression of plasma IL-6, IL-1ß, and TNF-ß, suggesting a link between TMAO and inflammation in ischemic HF through the TMAO/PKC/NF-κB signaling pathway [85]. Diabetes is known to make animal models more prone to IRI [86], and diabetes is positively associated with gut microbiota and TMAO imbalance [87]. In an IRI model induced by 30 min of ischemia followed by 2 h of reperfusion in mice, Dapagliflozin, an antidiabetic medication, reduced circulating TMAO levels and IRI-induced ferroptosis-mediated cardiomyocyte death by inhibiting MAPK signaling [88,89]. Accumulative evidence has demonstrated that a structural analog of choline, 3,3-dimethyl-1-butanol (DMB), inhibits TMA production in gut microbiota [90]. Rats undergoing permanent LAD ligation and subsequently treated with DMB exhibited improved cardiac function, hypertrophy, lung congestion, and left ventricular hemodynamics. Additionally, plasma TMAO and IL-8 levels were notably lower in DMB-treated MI rats, and there was a positive correlation between circulating TMAO levels and IL-8 in rats with MI, suggesting that TMAO may influence ischemic HF through IL-8 [91].

**Figure 2 ijms-26-02242-f002:**
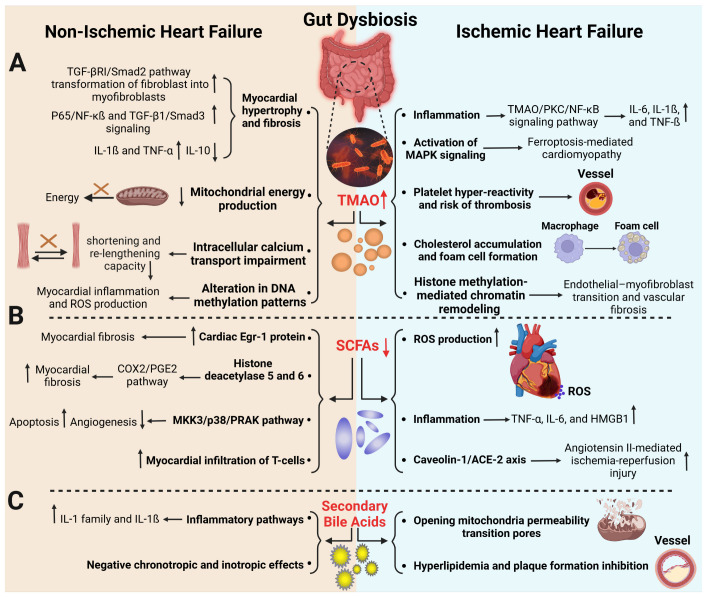
**Gut dysbiosis derivatives in experimental models of ischemic and non-ischemic HF.** (**A**) Following gut dysbiosis and increased TMAO, either myocardial hypertrophy and fibrosis through TGF-ß/Smad signaling pathways and altered DNA methylation pattern or myocardial dysfunction by calcium transport and energy production impairment can lead to non-ischemic HF. A high level of TMAO accelerates inflammatory pathways, platelet hyperactivity, cholesterol accumulation, and foam cell formation, which can all lead to thrombosis and clot formation in ischemic HF. Furthermore, TMAO primes MAPK signaling, leading to ferroptosis-mediated cardiomyopathy. In addition, histone methylation-mediated chromatin remodeling leading to endothelial–myofibroblast transition and vascular fibrosis results in ischemic HF. (**B**) Reduction in SCFAs after gut dysbiosis upregulates Egr-1 protein and T-cell myocardial infiltration, and enhances HDAC 5 and 6 activities that, through the MKK3/P38/PRAK pathway, causes less angiogenesis and more apoptosis, resulting in non-ischemic HF. Also, SCFAs decrement through enhancement in ROS and inflammatory cytokines production and C3/CAV-1/ACE-2 axis activation can lead to ischemic HF. (**C**) Changes in BAs, such as reduced deoxycholic acid and increased taurocholate, stimulate IL-1 and IL-1ß expression and worsen contractility, respectively, and affect mitochondrial apoptosis, leading to non-ischemic and ischemic HF through infarct expansion; in addition, with BAs reduction, cholesterol accumulates and plaque formation enhances and myocardium becomes prone to ischemic HF.

**Table 1 ijms-26-02242-t001:** Role of microbial metabolites in Ischemic and Non-ischemic Heart Failure.

Metabolite (Source)	Level Change	Mechanism of Action	Intermediate Effect	Heart Failure Impact	Ref.
**TMAO** (Choline, betaine, L-carnitine, lecithin, trimethyl lysine, γ-butyrobetaine, phosphatidylcholine, and **glycerophosphocholine-enriched diet [14,15]**)	↑TMAO	Increase IL-6, IL-1ß, and TNF-ß through TMAO/PKC/NF-κB signaling pathway	Inflammation	Ischemic HF	[85]
		Ferroptosis-mediated cardiomyocyte death by MAPK signaling	Cardiomyopathy and systolic dysfunction in IRI	Ischemic HF	[88,89]
		Upregulation of macrophage scavenger receptors CD36 and SR-A1 resulting in macrophage cholesterol accumulation and foam cell formation	Atherosclerosis progression	Ischemic HF	[92]
		Increase intracellular release of Ca2^+^ and platelet hyperactivity	Thrombosis	Ischemic HF	[93]
		Increase H3K4me3 and other H3 histone methylation in EC cells leading to EC reprograming into myofibroblast	Vascular fibrosis	Ischemic HF	[58]
		Activation of Smad3 signaling in cardiomyocyte	LV hypertrophy and fibrosis	Non-ischemic HF	[94]
		Transition of cardiac fibroblast to myofibroblast through the TGF-β/Smad3 signaling	Myocardial fibrosis	Non-ischemic HF	[61]
		Impair oxidative phosphorylation-dependent mitochondrial respiration leading to myocyte energy disturbance	Myocardial contractile dysfunction	Non-ischemic HF	[95]
		Inhibition of intracellular calcium removal leading to attenuation in cardiomyocyte contractility	Myocardial contraction dysfunction	Non-ischemic HF	[96]
		Activation p65/NF-κB and TGF-β1/Smad3 signaling pathways	Inflammation and myocardial fibrosis	Non-ischemic HF	[97]
		Increase in pro-inflammatory cytokines TNF-α and IL-1β and reduced IL-10	Inflammation and myocardial fibrosis	Non-ischemic HF	[98]
		Changing cardiomyocyte DNA methylation promoting inflammation, lipid metabolism, and oxidative stress	Inflammation and myocardial fibrosis	Non-ischemic HF	[99,100]
**SCFAs** (Dietary fibers and carbohydrates **[25,27]**)	↓Butyrate	Increase in oxidative stress and inflammation	Cardiomyocyte death in IRI	Ischemic HF	[101]
	↓Butyrate	Decrease in cardiomyocyte HMGB1 cytokine and antioxidant scavenger superoxide dismutase	ROS-mediated enhanced ischemia	Ischemic HF	[102]
	↓Propionate	Decrease in superoxide dismutase activity and silencing propionate/Caveolin-1/ACE-2 axis	Angiotensin-mediated exacerbated IRI	Ischemic HF	[103]
	↑Succinate	Oxidation by succinate dehydrogenase leading to ROS production	ROS-mediated enhanced ischemia	Ischemic HF	[104]
	↓Acetate	Upregulation of cardiac Egr1	Myocardial fibrosis	Non-ischemic HF	[105]
	↓Butyrate &↓Valproic acid	Histone deacetylation resulting in chromatin condensation and transcriptional silencing leading to decreased expression of COX2/PGE2 protein and atrial natriuretic peptide	Cardiac hypertrophy and fibrosis	Non-ischemic HF	[106,107,108,109]
	↓Butyrate	Downregulated MKK3/p38/PRAK pathway leading to decreased cardiac angiogenesis and increased myocyte apoptosis and oxidative stress	Inflammation and myocardial fibrosis	Non-ischemic HF	[110]
	↓Propionate	Increase in systemic inflammation and immune cells myocardial infiltration	Cardiac hypertrophy, fibrosis, and vascular dysfunction	Non-ischemic HF	[111]
**Bile acids** (Microbiota-mediated conversion of primary bile acids, which are synthesized in the liver and condensed in the gall bladder **[37,38]**)	↓DCA	TGR5-mediated increase in proinflammatory cytokines such as IL-1β	Enhanced myocardial fibrosis after MI	Ischemic HF	[112]
	↓UDCA	Downregulation of Akt and Bcl-2-associated death promoter phosphorylation results in translocation of Bcl-2-associated death promoter to mitochondria, which in turn promotes mPTP opening, enhancing mitochondrial damage	Cardiomyocyte apoptosis and enhanced reperfusion injury	Ischemic HF	[113]
	↓FXR-activation	Suppressing adiponectin secretion and silencing the AMPK-PGC-1α pathway leading to decreased mitochondrial biogenesis	Exacerbated inflammation and fibrosis after MI	Ischemic HF	[114]
	↓FXR-activation	Upregulation the expression of CD36 and ABCA1 in macrophages, leading to enhanced cholesterol uptake, promoting atherosclerosis and thrombotic events	Atherosclerosis and thrombosis	Ischemic HF	[115]
	↓TUDCA	Increase cardiac hypertrophy, fibrosis and apoptosis, and endoplasmic reticulum stress	Myocardial fibrosis and hypertrophy	Non-ischemic HF	[116]
	↑TC & ↑CA	Inhibition of intracellular cAMP, causing a negative chronotropic and ionotropic response in cardiomyocytes	Myocardial contraction dysfunction	Non-ischemic HF	[117,118,119]
**IPA** (**Tryptophan-enriched diet [45]**)	↓	Increased nicotinamide N-methyl transferase expression and decreased nicotinamide, NAD+/NADH, and SIRT3 levels	Myocardial fibrosis	Non-ischemic HF	[47]
**H_2_S** (**Cysteine-enriched diet [48,49]**)	↓	Increased Nox4 expression and mitochondrial ROS production leading to enhanced myocardial apoptosis and fibrosis	Myocardial hypertrophy and fibrosis	Non-ischemic HF	[120]
		Attenuation of myocardial nuclear factor E2-related factor 2 and the PI3K/AKT pathway activity resulting in enhanced oxidative stress	Myocardial hypertrophy and fibrosis	Non-ischemic HF	[121]
**PAGln** **(Protein-rich foods containing phenylalanine [52]**)	↑	Stimulating adrenergic receptors leading to reduced sympathetic-driven sarcomere contraction and myocyte function	Myocardial contraction dysfunction	Non-ischemic HF	[54]

Abbreviations: TMAO, trimethylamine N-oxide; HF, heart failure; IRI, ischemia reperfusion injury; EC, endothelial cell; SCFAs, short-chain fatty acid; HMGB1, high-mobility group box 1 protein; ROS, reactive oxygen species; ACE, angiotensin-converting enzyme; DCA, deoxycholic acid; UDCA, ursodeoxycholic; mPTP, mitochondrial permeability transition pores; FXR, farnesoid-X-receptor; TUDCA, Tauroursodeoxycholic acid; TC, taurocholate; CA, cholic acid; IPA, indole propionic acid; H_2_S, hydrogen sulfide; PAGln, phenylacetylglutamine.

TMAO was also demonstrated to not only act on the myocardium in ischemic HF, but also in the vasculature by modulating atherosclerosis. Wang et al. demonstrated that a high level of TMAO upregulates macrophage scavenger receptors CD36 and SR-A1, linking TMAO to macrophage cholesterol accumulation and foam cell formation, which promotes atherosclerosis [92]. Similarly, Koeth et al. depicted the TMAO as an atherogenic molecule as it results in overexpression of CD36 and scavenger receptor A in macrophages [122]. However, controversies regarding the role of TMAO in atherosclerosis exist, and TMAO was also found to inhibit endoplasmic reticulum (ER) stress in both cardiomyocytes and vascular smooth muscle cells, preventing heart and vasculature from fibrosis and atherosclerosis [123]. Zhu et al. demonstrated that increased TMAO plays a vital role in the intracellular release of Ca^2+^ and platelet hyperactivity in thrombotic events [93]. TMAO can induce chromatin modifications in endothelial cells in coronary arteries by increasing H3K4me3 and other H3 histone methylation marks, thus promoting endothelial-myofibroblast transition and vascular fibrosis through TMAO interaction with protein R-like endoplasmic reticulum kinase. Coronary vascular fibrosis hinders oxygen delivery to the myocardium and will prone it to ischemic HF. This supports the role of TMAO in the epigenetic regulation of ischemic HF [58].

#### 5.1.2. Non-Ischemic Heart Failure

Dysbiosis in non-ischemic HF patients has been shown to be accompanied by higher levels of TMAO in concomitance with myocardial fibrosis and myocardial dysfunction [11]. Dysbiosis in the direction of enhanced *Firmicutes* and *Proteobacteria* promotes TMA and TMAO production [13], and enrichment in these bacteria has been recognized in patients with HF [69,124]. Elevated fasting plasma levels of TMAO were found in stable HF patients compared to healthy controls and, furthermore, TMAO has been recognized as a potent all-cause mortality predictor independent of other traditional risk factors [125]. Similarly, plasma TMAO has been shown to predict long-term survival in HF patients, specifically when stratifying patients based on lower BNP levels has not demonstrated any prognosis utility. It was suggested that a combination of plasma TMAO and BNP levels may be considered for a more accurate prognosis prediction than BNP alone [126].

Several experimental studies have elucidated the role of increased plasma levels of TMAO in promoting adverse cardiac remodeling in non-ischemic HF models (Figure 2A and Table 1). Organ et al. observed elevated circulatory TMAO levels in mice who underwent transverse aortic constriction (TAC) surgery to induce pressure overload-HF compared to sham-operated animals [127]. Feeding mice a TMAO-supplemented diet before TAC promoted cardiac fibrosis and deteriorated cardiac function compared to mice fed a standard chow diet [127]. Similarly, mice receiving intraperitoneal TMAO after TAC had significantly higher TMAO plasma levels and enhanced LV hypertrophy and fibrosis compared to controls [94]. In this study, TMAO was demonstrated to exert its deleterious effects through the activation of Smad3 signaling [94]. TMAO also facilitates the transition of cardiac fibroblast to myofibroblast through TGF-β/Smad3 signaling, which is a hallmark of myocardial fibrosis [61]. TMAO was also shown to not only act as a pathological mediator but to negate the beneficial effects of exercise on cardiac dysfunction in mice fed a Western diet [128]. In this study, TMAO supplementation prevented exercise-induced improvement in cardiac fibrosis, concomitant with increasing pro-inflammatory TNF-α and decreasing anti-inflammatory IL-10 expression in the myocardium [128].

Regarding cardiomyocyte function, TMAO has been shown to impair oxidative phosphorylation-dependent mitochondrial respiration as well as β-oxidation, leading to myocyte energy disturbance, and myocardial dysfunction [95]. Additionally, TMAO induces lipopigment accumulation, known as glycogen aggregation, suggesting altered cellular energetic metabolism with a higher protein oxidative damage [96]. In the same study, TMAO also affected cardiomyocyte contractility by reducing fractional shortening through the inhibition of intracellular calcium removal [96].

Similar to the protective role of DMB in experimental models of the ischemic-HF model, mice receiving DMB supplementation in their drinking water in the non-ischemic HF model induced by TAC also exhibited lower levels of circulating TMAO together with attenuated cardiac hypertrophy, fibrosis, arrhythmias, and inflammation [97]. The study demonstrated that suppressing TMAO level by DMB was associated with inhibition of the p65/NF-κß and TGF-β1/Smad3 signaling pathways [97]. Similarly, DMB treatment mitigated the deleterious effects of the Western diet on cardiac dysfunction. Western diet-fed mice showed higher plasma TMAO, and worse systolic and diastolic function compared to controls, as well as elevated pro-inflammatory cytokines TNF-α and IL-1β and reduced IL-10, all of which were mitigated by DMB treatment [98]. Gut microbiota metabolism of choline influences DNA methylation across multiple organs, including the heart and colon. In a gnotobiotic mouse model, bacterial choline consumption altered DNA methylation patterns. These changes may result in gene hypermethylation or hypomethylation, affecting inflammation, lipid metabolism, and oxidative stress, eventually resulting in non-ischemic HF [99,100].

Not all TMAO roles in HF are harmful. TMAO, naturally being an osmotic substance, protects proteins to be structurally stabilized during hydrostatic pressure [129], facilitating cell adaptation during the volume overload in HF [130,131]. Therefore, it has been hypothesized that TMAO quantity enhancement is a compensatory response, similar to releasing natriuretic peptides, to protect cardiomyocytes against hydrostatic pressure [75].

### 5.2. SCFAs

#### 5.2.1. Ischemic Heart Failure

Clinical evidence has shown that SCFAs play a role in ischemic HF. Liu et al. demonstrated a change in both microbiota composition and SCFAs levels in patients suffering from acute MI. Fecal sample analyses showed that MI patients had lower acetate and butyrate levels, and higher propionate and isovalerate levels compared to controls [132]. However, propionate levels in preoperative stool were shown to negatively correlate with the degree of myocardial injury based on troponin I plasma levels in patients undergoing cardiopulmonary bypass surgery [103]. Elevated acetate levels and reduced butyrate and propionate levels have been identified as distinguishing metabolites between patients with acute MI and healthy controls. Moreover, the acetate/propionate ratio was shown to be the best predictor of survival in HF patients [82].

In agreement with clinical studies showing reduced butyrate and propionate levels in patients with acute MI compared to healthy controls [132], experimental studies have elucidated protective roles of replenishing SCFAs against IRI (Figure 2B and Table 1). Several studies have shown that butyrate or propionate supplements reduce myocardial infarct size upon IRI. Rats fed with a diet supplemented with butyrate four weeks prior to IRI exhibited significantly smaller infarct size, less cardiac oxidative stress and inflammation, as well as significantly less cardiomyocyte apoptosis compared to rats that did not receive butyrate [101]. Similarly, Hu et al. demonstrated that preconditioning with systemic butyrate injections 30 min before the onset of ischemia decreased IRI-induced infarct size and the expression of pro-inflammatory high-mobility group box 1 protein (HMGB1) cytokine in the cardiomyocytes, while enhancing the expression of antioxidant scavenger superoxide dismutase [102]. The supplementation of propionate in the drinking water also reduced infarct size, enhanced cardiac superoxide dismutase activity, and improved cardiac function after AngII-aggravated IRI in mice [103]. It was hypothesized that propionate inhibits angiotensin-converting enzyme through G-protein-coupled receptor 41 (GPR41) activation, thus limiting myocardial IRI through the propionate/Caveolin-1/ACE-2 axis [103].

In contrast to the protective effects of butyrate and propionate, succinate, an intermediate metabolite in SCFAs fermentation, accumulates in the cell during ischemia and is subsequently oxidized by succinate dehydrogenase during reperfusion. This process is a major driver for ROS production [104]. Recent studies have shown that the inhibition of succinate dehydrogenase by administering the prodrug dimethyl malonate (DMM) during ischemia upon LAD ligation prevents succinate accumulation, and oxidation thus attenuating infarct size in myocardial IRI mice. Additionally, rapidly hydrolysable forms of DMM were able to inhibit succinate oxidation and mitochondrial ROS production and therefore decrease infarct size when administered during the therapeutic window of reperfusion in mice [133,134].

#### 5.2.2. Non-Ischemic Heart Failure

Stable non-ischemic HF patients exhibited lower plasma concentrations of propionate, butyrate, and isovalerate compared to healthy controls, whereas no difference was found in levels of acetate and valerate [135].

Experimental models of non-ischemic HF have shown that SCFAs affect several aspects of the disease pathophysiology (Figure 2B and Table 1). Hypertensive mice, induced by the DOCA-salt model [136], when fed a high-fiber diet or if the drinking water was supplemented with acetate, exhibited significantly reduced cardiac hypertrophy and fibrosis, which led to improved cardiac function [105]. Transcriptome analysis linked the protective effects of fiber and acetate to the downregulation of cardiac Egr1, a key regulator of myocardial fibrosis [105]. TAC-induced cardiac fibrosis was compared in wild-type (WT) mice and gut microbiota-depleted mice treated with antibiotics, which exhibited dysbiosis and reduced levels of microbial metabolites, including SCFAs. The enhanced cardiac hypertrophy and fibrosis as well as ECM dysfunction in microbiota-depleted mice were attenuated by supplementing antibiotics with acetate and propionate. It was shown that the protective effects of acetate and propionate were mediated via GPR41 and GPR43 receptors in cardiac fibroblasts [137]. SCFAs have been found to influence the function of ten–eleven translocation methylcytosine dioxygenase enzymes, which are crucial for DNA demethylation [106]. SCFAs are recognized as key inhibitors of histone deacetylases (HDACs), preventing the removal of acetyl groups from histone lysines. This inhibition prevents chromatin condensation and transcriptional silencing [107]. Thereby, HDACs participate in cardiac hypertrophic responses and SCFAs are effective in HDAC-mediated cardiac fibrosis and hypertrophy [107]. Also, valproic acid, another SCFA, inhibits histone acetylation, and, in response, the acetylation of the mineralocorticoid receptor of cardiomyocytes increases, leading to decreased cardiac hypertrophy and fibrosis in hypertensive rats [108]. Butyrate was shown to be effective in mitigating cardiac fibrosis in an Ang II-induced cardiac fibrosis model. Rats supplemented with butyrate exhibited reduced cardiac fibrosis, and improved hypertrophic markers compared to hypertensive rats that did not receive butyrate. In this study, the protective action of butyrate was associated with reduced HDAC 5 and 6 activities, the decreased transcript and protein expression of COX2/PGE2, and the decreased expression of atrial natriuretic peptide, both in transcript and protein levels [109]. Butyrate supplementation has also been shown to reduce metabolic syndrome symptoms in mice on a high-fat diet, decrease myocyte apoptosis and oxidative stress, and improve cardiac angiogenesis. These effects of butyrate were mediated through the MKK3/p38/PRAK pathway [110]. A similar experimental approach was used to evaluate the efficacy of propionate against non-ischemic HF in an Ang II-induced hypertensive model using Apo E-KO mice. Propionate significantly improved cardiac hypertrophy, fibrosis, vascular dysfunction, and hypertension compared to controls. Propionate exerted its protective effects by reducing systemic inflammation, evidenced by decreased splenic effector memory T cell and T helper 17 frequencies, and by limiting local immune cell infiltration in the heart [111].

### 5.3. Bile Acids

#### 5.3.1. Ischemic Heart Failure

Alterations in BAs levels have been reported in the plasma of ischemic HF patients. Decreased levels of primary BAs and an increased ratio of secondary to primary BA in the plasma of chronic HF patients have been reported [12]. Additionally, patients with acute MI exhibited decreased plasma levels of deoxycholic acid (DCA), glycoursodeoxycholic acid, and ursodeoxycholic (UDCA) acid compared to healthy controls [112].

Experimental studies have also reported protective roles for BAs in ischemic HF (Figure 2C and Table 1). UDCA administration before global ischemia in Langendorff-perfused rat hearts resulted in improved coronary flow and contractility, together with lower levels of myocardial damage marker lactate dehydrogenase [138]. Similarly, in the in vivo model of IRI, UDCA administration prior to coronary ligation significantly reduced myocardial infarct size in rats. Mechanistically, UDCA was shown to activate Akt and Bcl-2-associated death promoter phosphorylation, preventing translocation of Bcl-2-associated death promoter to mitochondria, which in turn inhibited mPTP opening, reducing mitochondrial damage and myocyte apoptosis during reperfusion [113]. A similar study showed protective action of DCA in mice receiving gavage of DCA prior to permanent LAD ligation by significantly better outcomes characterized by improved cardiac function, smaller infarct size, less myocardial fibrosis, and lower expression of IL-1β expression in the heart, which was dependent on bile acid receptor TGR5 [112]. BAs may also bind to farnesoid-X-receptor (FXR), a nuclear receptor that mainly functions as a sensor for bile acids. There is some discrepancy in the role of FXR activation in myocardial function upon IRI. Increased cardiac FXR expression using FXR agonists promoted mitochondrial apoptotic responses, including the opening of mPTP, mitochondrial potential loss, cytochrome c release, and caspase-9/-3 activation [139]. Accordingly, pharmacological FXR inhibition in both FXR-KO mice and WT mice reduced infarct size and myocardial apoptosis compared to their respective controls [139]. In contrast, a protective role for FXR signaling against myocardial ischemia was also shown. After permanent LAD ligation, mice who received gavage of synthetic FXR agonist GW4064, had markedly improved EF, enhanced mitochondrial biogenesis, and reduced cardiomyocyte cell death together with less inflammation and fibrosis in the infarcted myocardium [114]. Further evaluation of downstream molecules in FXR activation showed that enhanced adiponectin secretion and the activation of the AMPK-PGC-1α pathway may protect cardiomyocytes against hypoxic injury [114]. Discrepancies in the effects of FXR activation may be due to its post-translational modification status. SUMOylation of FXR was found to be protective against myocardial IRI, since mice expressing a SUMOylation-resistant FXR had significantly increased levels of myocardial apoptosis and larger infarct size as compared to mice expressing WT FXR [140]. Decreased FXR SUMOylation was also found to promote mitochondrial dysfunction and swelling in the cardiomyocytes [140].

BAs not only directly act on the myocardium, but also affect ischemic HF progression by regulating cholesterol metabolism, atherogenic plaque, and thrombus formation [141]. FXR–ligand interaction was shown to ameliorate hyperlipidemia and plaque formation via two mechanisms. In Apo lipoprotein E-knock out mice, FXR activation by BAs or synthetic ligands inhibited the expression of sterol regulatory element binding protein-1c in the liver, thus promoting cholesterol and triglyceride breakdown in the liver, which subsequently prevents hyperlipidemia. Additionally, FXR activation downregulates the expression of CD36 and ABCA1 in macrophages, leading to decreased cholesterol uptake, thus preventing atherosclerosis [115].

#### 5.3.2. Non-Ischemic Heart Failure

Altered BA levels have recently been reported in the serum of HF patients caused by metabolic dysfunction-associated fatty liver disease. In this study, lower levels of ursodeoxycholic acid and hyocholic acid were observed in HF patients and these lower levels were positively correlated to HF status [142].

BAs have been shown to play dual roles in non-ischemic HF (Figure 2C and Table 1). Tauroursodeoxycholic acid (TUDCA) was found to counteract obesity-induced cardiomyopathy-mediated HF in mice. TUDCA supplementation in the diet of obese mice improved cardiac function and attenuated cardiac hypertrophy, fibrosis and apoptosis, and ER stress in cardiomyocytes [116]. However, BAs have also been attributed to negative inotropic and chronotropic properties. Taurocholate (TC) has been shown as an agonist for the muscarinic-2 receptor in cardiomyocytes, which inhibits the intracellular cAMP, causing a negative chronotropic response in neonatal rat cardiomyocytes [117] and reduced contractility in rat cardiomyocytes [118]. Similar findings were reported for the effect of cholic acid as a negative chronotropic regulator of cardiomyocytes [119].

### 5.4. Role of Less-Studied Microbial Metabolites in HF

Beyond TMAO, SCFAs, and BAs, other microbial products such as IPA, H_2_S, and PAGln have also been implicated in both ischemic and non-ischemic HF (Table 1) [47,54,143,144]. Comparing the plasma level of IPA between HF patients with healthy age- and sex-matched controls showed decreased IPA plasma level in HF patients [47]. A similar pattern of decreased IPA was shown in the plasma of mice with HF induced by a high-fat diet and the inhibition of nitric oxide. Exploring the downstream pathway showed IPA suppresses the expression of nicotinamide N-methyl transferase, restored nicotinamide, NAD+/NADH, and SIRT3 levels, leading to enhanced cardiac function including improved LVEF, E/A ratio, E/e′ ratio, heart weight, and LV mass/surface area [47]. Zhu et al. highlighted the protective effects of H_2_S in ischemic HF, showing that pre-treatment with NaHS significantly reduced infarct size and alleviated systolic dysfunction in rats following permanent LAD ligation [143]. S-propargyl-cysteine, a new regulator of endogenous H_2_S, protects against MI and ischemic HF by mitigating oxidative stress damage through enhanced cystathionine-γ-lyase activity and elevated plasma H_2_S levels [120]. H_2_S has been shown to enhance cardiac function and decrease myocardial apoptosis in a rat model of isoproterenol-induced cardiac hypertrophy by lowering Nox4 expression and reducing mitochondrial ROS production [144]. In mice, treatment with sodium sulfide (Na_2_S) decreases cardiac hypertrophy and left ventricular dilation, while enhancing left ventricular function after ischemic HF induction, with these effects being dependent on thioredoxin 1 [145]. A diet rich in fresh garlic leads to increased H_2_S levels, which can activate myocardial nuclear factor E2-related factor 2 via the PI3K/AKT pathway. This activation reduces cardiac hypertrophy and oxidative stress by enhancing the antioxidant defense system in insulin-resistant rats fed fructose [121]. An analysis of US population revealed that HF patients with reduced and preserved LVEF exhibited elevated plasma PAGln levels compared to those without HF. This observation was consistent with findings from the European population, where plasma PAGln levels were also higher in HF patients than in non-HF individuals. In addition, the plasma level of PAGln was negatively and positively correlated with LVEF and BNP, respectively, supporting the association between PAGln and HF severity [54]. Investigating downstream pathways through in vitro and in vivo experiments revealed that PAGln, by activating adrenergic receptors, reduces sympathetic-driven sarcomere contraction and myocyte function. This finding highlights PAGln’s role as a negative inotropic agent, contributing to myocardial dysfunction and heart failure [54].

## 6. Therapeutic Perspective of the Gut–Heart Axis in Ischemic and Non-Ischemic HF

The gut microbiota is an attractive therapeutic target as modulating the gut microbiome could be implemented easily by dietary interventions, probiotics, prebiotics, or even fecal microbiota transplantation (FMT). Together with the ease of access and modest cost of interventions, the concept of gut microbiota modulation in CVD has been emerging [146,147]. Probiotics, prebiotics, dietary interventions, and FMT thus may provide new therapeutic avenues for managing both ischemic and non-ischemic HF. Several studies have demonstrated changes in gut microbiota metabolites upon treatment with probiotics, prebiotics, dietary interventions, and FMT in the context of HF.

### 6.1. Dietary Interventions

Dietary interventions have been indicated to be another means of modulating gut microbiota composition and dynamics. A Western diet, low in fiber and high in saturated fats and refined carbohydrates, can decrease the abundance of bacteria-producing SCFAs and increase TMAO and inflammatory cytokines arising from a variety of health issues including HF [148,149]. Conversely, the Mediterranean diet, a diet rich in fiber and unsaturated fatty acids, can positively alter the microbial intestinal composition by shifting the microbiome towards the beneficial bacteria *Prevotella* and *Bacteroides* while shifting away from *Firmicutes* [149]. Adherence to the Mediterranean diet brings positive health associations, including producing SCFAs and anti-inflammatory properties, reducing the risk of inflammation-associated diseases [149]. The Mediterranean diet has also been associated with lower TMAO levels, partly due to the presence of DMB, a compound that inhibits TMAO production. Notably, DMB is found in extra virgin olive oil, a key element of this heart-healthy dietary pattern [150]. The MEDIT-AHF study, which tracked patients with acute HF, found that while adherence to a Mediterranean diet did not lower long-term mortality, it was linked to a reduced risk of rehospitalization [151]. An in vivo experiment showed mice fed with a plant-based diet compared to mice on a chow diet had a notable increase in SCFAs-producing bacteria comprising *Bacteroides* and *Alloprevotella*, accompanied by a significant decrease in *Porphyromonadaceae* and *Erysipelotrichaceae* [152].

### 6.2. Probiotics

Probiotics are living microorganisms, such as bacteria and yeasts, which confer health benefits to the host when administered in adequate quantities in a viable state [153]. Probiotics have been shown to reduce systemic inflammation, which is a key factor in both ischemic and non-ischemic HF [154]. For instance, in a randomized controlled trial, Pourrajab et al. have demonstrated that probiotic yogurt consumption compared to regular yogurt in patients with chronic HF resulted in a significant increase in serum levels of the anti-inflammatory molecule soluble tumor necrosis factor-like weak inducer of apoptosis (sTWEAK), suggesting that probiotics may be useful in improving the inflammatory status in HF patients [155].

Probiotics have also demonstrated potential in promoting myocardial repair as both preventative as well as therapeutic regimen [156,157]. Antibiotic-treated mice exhibited significantly depleted gut microbiota and reduced levels of acetate, butyrate, and propionate in both serum and fecal samples compared to untreated mice. These mice also showed a sharply reduced survival rate following MI. Supplementing antibiotic-treated mice with *Lactobacillus* probiotics after MI restored gut microbiota composition and increased propionate levels in serum and fecal samples [157]. Additionally, probiotics supplementation enhanced myocardial repair by increasing myeloid cell infiltration, specifically CX3CR1^+^ monocytes, to the infarct site and significantly improved post-MI survival rates [157]. Similarly, oral administration of a newly engineered probiotic cocktail containing *Escherichia coli Nissle* before IRI has effectively maintained steady plasma levels of the SCFAs propionate and butyrate. This engineered probiotic reduced inflammation by suppressing the NF-κB pathway, decreased neutrophil infiltration at the infarct site, and promoted wound healing by facilitating macrophage anti-inflammatory polarization, helping to preserve myocardial function after IRI [156].

### 6.3. Prebiotics

Prebiotics are substrates utilized explicitly by host microorganisms to restore gut microbiota composition. All molecules presently classified as prebiotics fall into one of two categories: carbohydrates that can be metabolized by the gut flora or fermentable dietary fiber [153]. As such, prebiotics may significantly affect cardiovascular outcomes by increasing the production of SCFAs. Indeed, a high-fiber diet was shown to be as effective as dietary supplementation of acetate alone in attenuating blood pressure, cardiac hypertrophy, and cardiac fibrosis, as well as improving cardiac function in DOCA-salt hypertensive mice [105].

### 6.4. FMT

FMT involves administering fecal material from a healthy donor into the intestinal tract of a recipient [158]. FMT is a traditional treatment for gastrointestinal disorders, but its role in CVD treatment is not fully understood [158]. FMT could help restore gut flora balance in patients with HF, particularly those who have undergone intensive antibiotic therapy or have significant gut dysbiosis [8]. In a study by Hatahet et al., FMT treatment of pre-HF obese mice led to an increase in circulating levels of the SCFA butyrate, and FMT increased the levels of butyrate-producing bacteria *Lactobacillus* [159]. FMT in obese mice improved diastolic dysfunction and cardiac hypertrophy suggesting that FMT’s positive effect on HF development could occur due to SCFA butyrate [159] (136). Doxorubicin (DOX) was previously indicated to alter gut microbiota composition and their metabolites [160], which is concomitant with causing myocardial mitochondrial dysfunction. In mice with DOX-induced cardiotoxicity, FMT restored gut dysbiosis and serum levels of metabolite indole-3-propionic acid. In cardiomyocytes, indole-3-propionic acid facilitated the translocation of transcription factor Nfe2l2 from the cytoplasm to the nucleus, thereby activating the expression of antioxidant molecules, reducing ROS production, and inhibiting excessive mitochondrial fission. Thereby, FMT was proposed as a promising therapy for preventing DOX-induced HF [161].

## 7. Conclusions, Limitations, and Future Perspectives

The role of gut microbiota extends beyond the gastrointestinal tract, interacting with distal organs such as the cardiovascular system, a relationship termed the gut–heart axis. This interaction is driven by metabolites such as TMAO, SCFAs, BAs, IPA, H_2_S, and PAGln. Dysbiosis, or alterations in gut microbiota composition, has been reported in patients suffering from both ischemic and non-ischemic HF. Additionally, alterations in circulating TMAO, SCFAs, BAs, IPA, H_2_S, and PAGln have been reported in HF patients. Furthermore, experimental animal models have demonstrated that these metabolites can regulate multiple aspects of HF pathophysiology. 

A limitation of this narrative review is that we have not applied a systematic search approach, which may possibly introduce selection bias in the studies covered. We mainly focused on the roles of TMAO, SCFAs, BAs, IPA, H_2_S, and PAGln underlying HF. However, gut microbes secrete several other metabolites which may play a role in HF pathophysiology. Lastly, animal models of ischemic and non-ischemic HF are reductionist and limited and do not fully represent the complex multifactorial etiology and pathophysiology of HF as observed in patients. As such, there is a translational limitation in the roles gut microbiota metabolites have been attributed in distinct molecular pathways in animal models of HF. 

Findings on the role of these metabolites in ischemic and non-ischemic HF opens up a novel additional perspective toward understanding HF as a systemic condition, not just a cardiac-centric issue. Also, future research should focus on whether microbial metabolites play roles in different stages of HF pathophysiology. Additionally, microbial metabolites can affect a variety of cells, both in the heart and in tissues such as the vasculature. These distinct cell-specific roles in HF remain to be elucidated. This emerging field of research opens new avenues for HF therapies by restoring gut microbiota through prebiotics, probiotics, dietary interventions, or FMT and, as such, by normalizing circulating TMAO, SCFAs, BAs, IPA, H_2_S, and PAGln to their levels in healthy subjects. However, future research is needed to assess the safety and efficacy of these microbiome-based interventions as well as possible interactions or synergistic benefits with existing guideline-based HF pharmacotherapies, aiming to create comprehensive treatment plans that leverage the gut–heart axis.

## Figures and Tables

**Figure 1 ijms-26-02242-f001:**
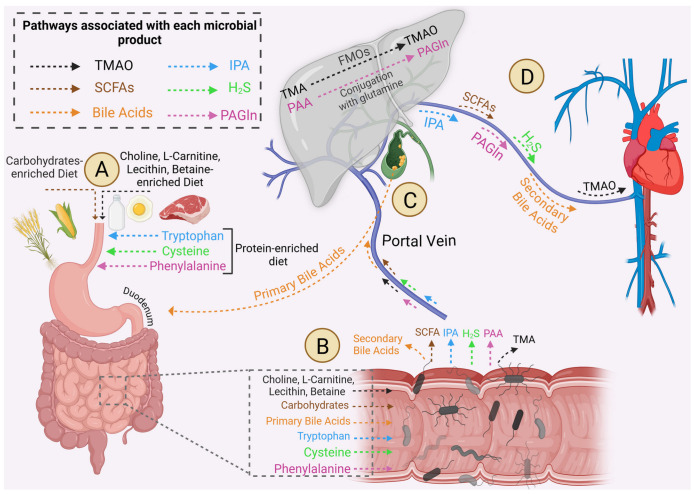
**Microbiota metabolites in the gut–heart axis.** (**A**) After consuming a regular diet, gastrointestinal enzymes take them apart into micronutrients such as betaine, L-carnitine, phosphatidylcholine, tryptophan, cysteine, phenylalanine, and non-digestible carbohydrates, including fibers such as inulin, pectin, and resistant starch. (**B**) The gut microbiota converts food-derived compounds into TMA, SCFAs, IPA, H_2_S, and phenylacetic acid (PAA). Also, it changes the duodenum-released primary bile acids into secondary bile acids. (**C**,**D**) They are reabsorbed into the portal vein and enter the liver. Flavin-containing Monooxygenase (FMO) transforms TMA into TMAO and releases it into the hepatic vein. Hepatocytes and enterocytes consume most SCFAs and IPA to tighten their intercellular junction and maintain intestinal integrity; the rest are released into the systemic circulation. In addition, secondary bile acids are either reabsorbed by the liver and go back to enterohepatic circulation or enter the systemic circulation to ultimately affect the heart. Microbiota-driven H_2_S regulates inflammation and tissue repair within the GI tract and as released circular gasotransmitter facilitates vasodilation and other systemic effects. Liver PAA conjugation with glutamine results in PAGln production and secretion into the portal vein and subsequently in systemic circulation.

## Data Availability

Not applicable.

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
