# Peer review of "Role of Gut Microbial Metabolites in Ischemic and Non-Ischemic Heart Failure"

_ijms, 2025, doi:10.3390/ijms26052242_

Round 1

Reviewer 1 Report

Comments and Suggestions for Authors

Author Response

Comment 1: There are some conceptual errors in this manuscript. For example: (1) On page 7, lines 218-235, the authors mentioned that an increased plasma level of TMAO has been demonstrated in the chronic myocardial ischemia model. Subsequently, the authors provided numerous examples based on an animal model with a permanent LAD ligation for 4-6 weeks to support this point. However, the concept of chronic myocardial ischemia refers to the consequence of slow coronary artery occlusion. The models in these studies were used to investigate the long-term changes following acute myocardial ischemia, not for chronic myocardial ischemia.

(2) On page 8, lines 259-269, the authors attempted to use the TAMO changes in HFpEF patients to represent the TAMO changes in non-ischemic heart failure patients. However, the pathogenesis and mechanism of HFpEF are complex. Multiple factors can lead to HFpEF. Ischemic heart failure and HFpEF are not mutually exclusive. In fact, some scholars think that ischemia caused by coronary microvascular dysfunction is a major cause of HFpEF. Obstructive coronary arteries are not the only cause of ischemic heart failure. The following reviews may be helpful in understanding this issue.

https://pubmed.ncbi.nlm.nih.gov/36423427/

https://pubmed.ncbi.nlm.nih.gov/30684452/

Due to these conceptual problems, the authors may need to reconsider the structure of this paper.

Response 1: 1. We agree with the Reviewer that permanent LAD ligation initially causes acute ischemia. Since the ensuing cardiac remodeling is long-term, we named this “chronic ischemia”, which may have caused an inaccurate interpretation. Therefore, for clarity, we removed words “chronic” and “acute” from experimental models of myocardial ischemia and explained better the differences between the two models of permanent LAD ligation model for several weeks with short ligation followed by reperfusion model known as IRI. For the clinical study we only used term “chronic” or “acute” if it was specifically mentioned in the reference that we used.

  1. We acknowledge the complex and multifactorial nature of HFpEF. In light of the reviewer's fair point regarding ischemia and microvascular dysfunction as HFpEF etiologies, we have decided to remove the HFpEF studies from the non-ischemic HF sections throughout the manuscript.

Comment 2: In this review, the authors enumerated too many specific research findings but did not demonstrate the authors' in-depth thinking about these research results.

Response 2: We have now elaborated on our in-depth thinking regarding the role of gut microbial metabolites in heart failure in section 7, in the conclusion and future perspectives section.

Reviewer 2 Report

Comments and Suggestions for Authors

The review is very interested and could be published after authors make some appropriate changes in the manuscript.

1. Title- This is a "narrative review" and this should be included in the title

2. There are other important metabolites such as Indole Propionic Acid, Hydrogen sulfide, Phenylacetylglutamine. Please discuss in depth the molecular mechanism of those too.

3. It would be nice to present in a table a summary of the source, mechanism of action and HF impact of gut metabolites 

4. Please discuss dietary modifications to improve and maintain gut microbial dynamics

5. Discuss limitations of your study

6. Discuss future perspectives 

Author Response

Responses to the Reviewer 2:

Comment 1: This is a "narrative review" and this should be included in the title.

Response 1: We have now added ‘narrative review’ to the title.

Comment 2: There are other important metabolites such as Indole Propionic Acid, Hydrogen sulfide, Phenylacetylglutamine. Please discuss in depth the molecular mechanism of those too.

Response 2: We have now expanded the metabolites discussed with studies on Indole Propionic Acid, Hydrogen sulfide, Phenylacetylglutamine and have added two major paragraphs:

  • Section 2 explaining how these metabolites are produced: lines 136-150
  • Section 5.4. explaining how IPA, H2S, and PAGln can affect ischemic and non-ischemic HF: lines 503-536

Comment 3: It would be nice to present in a table a summary of the source, mechanism of action and HF impact of gut metabolites. 

Response 3: We summarized all data related to metabolites in a new Table 1.

Comment 4: Please discuss dietary modifications to improve and maintain gut microbial dynamics.

Response 4: We have now discussed dietary modifications on gut microbiota dynamics in HF in section 6.

Comment 5: Discuss limitations of your study.

Response 5: We have now added section 7, discussing the limitations of a narrative review, the roles additional gut microbial metabolites may play, and the limitations of translation from animal models to complex clinical presentation of heart failure.

Comment 6: Discuss future perspectives. 

Response 6: We proposed some recommendations for future research in section 7 regarding the roles of gut microbiota metabolites in HF in a cell-specific manner, in different stages of HF progression, and the knowledge gap in determining the efficacy of microbial modulating approaches in heart failure.

Reviewer 3 Report

Comments and Suggestions for Authors

The authors have described the role of gut microbial metabolites in ischemic and non-ischemic heart failure is increasingly recognized as a critical factor in the pathophysiology and progression of these conditions. Gut microbial metabolites, such as TMAO, SCFAs, and secondary BA, influence cardiovascular health through various mechanisms, including inflammation, and oxidative stress. etc,

Although it is easy to understand the narrative nature of the text, its scientific value would have been significantly enhanced if the review had been conducted more systematically. A structured approach, following established methodologies for literature review, would have ensured greater rigor, reproducibility, and comprehensiveness in the analysis. This would have strengthened the validity of the conclusions and provided a more robust foundation for further research.

The diagrams are excellent and explain it well.

The authors should also discuss the role of gut microbial metabolites in epigenetic regulation in ischemic and non-ischemic heart failure.

Author Response

Comment 1: Although it is easy to understand the narrative nature of the text, its scientific value would have been significantly enhanced if the review had been conducted more systematically. A structured approach, following established methodologies for literature review, would have ensured greater rigor, reproducibility, and comprehensiveness in the analysis. This would have strengthened the validity of the conclusions and provided a more robust foundation for further research.

Response 1: We have now discussed this limitation in section 7.

Comment 2: The authors should also discuss the role of gut microbial metabolites in epigenetic regulation in ischemic and non-ischemic heart failure.

Response 2: We have now discussed the role of epigenetic regulation by gut microbial metabolites. We provide a brief description on the role of epigenetic regulation in HF pathophysiology in section 3 in lines 180-186. Then we provide explanation about the role of gut microbial metabolites in epigenetic changes during HF pathophysiology. The role of TMAO in epigenetic changes of ischemic HF is discussed in lines 297-302. Role of TMAO in epigenetic changes of non-ischemic HF model: is discussed in lines 347-352. We previously have discussed the role of SCFA Butyrate in histone deacetylation (lines 384-388 in previous version) and now we elaborate more on this topic in lines 415-421. Also we mentioned to the role valporic acid as well in lines 421-424. To date, no original data on the role of secondary bile acids in epigenetic regulation of heart failure has been published.

Round 2

Reviewer 2 Report

Comments and Suggestions for Authors

Thank you for addressing all comments